# Connexin43, A Promising Target to Reduce Cardiac Arrhythmia Burden in Pulmonary Arterial Hypertension

**DOI:** 10.3390/ijms25063275

**Published:** 2024-03-14

**Authors:** Matus Sykora, Barbara Szeiffova Bacova, Katarina Andelova, Tamara Egan Benova, Adriana Martiskova, Lin-Hai Kurahara, Katsuya Hirano, Narcis Tribulova

**Affiliations:** 1Centre of Experimental Medicine, Institute for Heart Research, Slovak Academy of Sciences, 84104 Bratislava, Slovakia; matus.sykora@savba.sk (M.S.); barbara.bacova@savba.sk (B.S.B.); katarina.andelova@savba.sk (K.A.); tamara.benova@savba.sk (T.E.B.); adriana.martiskova@savba.sk (A.M.); 2Department of Cardiovascular Physiology, Faculty of Medicine, Kagawa University, Takamatsu 761-0793, Japan; hailin@med.kagawa-u.ac.jp (L.-H.K.); hirano.katsuya@kagawa-u.ac.jp (K.H.)

**Keywords:** PAH, myocardial hypertrophy and fibrosis, aberrant connexin-43, cardiac arrhythmias

## Abstract

While essential hypertension (HTN) is very prevalent, pulmonary arterial hypertension (PAH) is very rare in the general population. However, due to progressive heart failure, prognoses and survival rates are much worse in PAH. Patients with PAH are at a higher risk of developing supraventricular arrhythmias and malignant ventricular arrhythmias. The latter underlie sudden cardiac death regardless of the mechanical cardiac dysfunction. Systemic chronic inflammation and oxidative stress are causal factors that increase the risk of the occurrence of cardiac arrhythmias in hypertension. These stressful factors contribute to endothelial dysfunction and arterial pressure overload, resulting in the development of cardiac pro-arrhythmic conditions, including myocardial structural, ion channel and connexin43 (Cx43) channel remodeling and their dysfunction. Myocardial fibrosis appears to be a crucial proarrhythmic substrate linked with myocardial electrical instability due to the downregulation and abnormal topology of electrical coupling protein Cx43. Furthermore, these conditions promote ventricular mechanical dysfunction and heart failure. The treatment algorithm in HTN is superior to PAH, likely due to the paucity of comprehensive pathomechanisms and causal factors for a multitargeted approach in PAH. The intention of this review is to provide information regarding the role of Cx43 in the development of cardiac arrhythmias in hypertensive heart disease. Furthermore, information on the progress of therapy in terms of its cardioprotective and potentially antiarrhythmic effects is included. Specifically, the benefits of sodium glucose co-transporter inhibitors (SGLT2i), as well as sotatercept, pirfenidone, ranolazine, nintedanib, mirabegron and melatonin are discussed. Discovering novel therapeutic and antiarrhythmic strategies may be challenging for further research. Undoubtedly, such research should include protection of the heart from inflammation and oxidative stress, as these are primary pro-arrhythmic factors that jeopardize cardiac Cx43 homeostasis, the integrity of intercalated disk and extracellular matrix, and, thereby, heart function.

## 1. Introduction

PAH is a rare but progressive and potentially life-threatening cardiovascular disorder of various etiology, with a female predominance and increased male severity [1,2]. Mean pulmonary arterial pressure >20 mmHg and pulmonary capillary wedge pressure ≤14 mmHg are relevant diagnostic markers of this disease [3]. PAH development is promoted by disturbed signaling of the transforming growth factor-β (TGFβ) family and gene mutations of the bone morphogenetic protein receptor 2 (BMPR2) [4,5,6]. Sex hormones can determine the expression of receptors (including BMPR2), ligands and antagonists within the TGFβ family in a sex-specific manner [1]. Beyond this, sex hormones are differently associated with function of the right ventricle (RV) in male and female PAH patients. There is an interplay of sex hormones and long-term RV adaptation in PAH patients [7].

The pathophysiology of PAH is characterized by endothelial dysfunction, proliferation of smooth muscle cells and vasoconstriction, leading to progressive remodeling of the pulmonary arterial tree [2,8]. Endothelial cells, smooth muscle, and fibroblast, as well as inflammatory cells and platelets, may be implicated in the increase of pulmonary vascular resistance leading to increased RV afterload and RV heart failure (HF) over time [9,10,11,12]. Notably, inflammation and cytokines that are crucial for the regulation of immune responses have been involved in the pathogenesis of PAH [13].

The length of survival has improved with recent advances in specific therapy [5,11], depending on the restoration of RV function [12]. Nevertheless, sudden cardiac death (SCD) associated with malignant arrhythmias and RV failure accounts for approximately 30% of PAH-related deaths [14,15]. Due to its non-specific symptoms, PAH is often diagnosed late and at an advanced stage, which jeopardizes RV function and increases the risk of arrhythmias. Therefore, besides transthoracic echocardiography, the knowledge of typical ECG signs and the use of magnetic resonance imaging (MRI), as well as dual/energy computed tomography (CT), single photon emission CT and ventilation perfusion scans, could help to detect PAH earlier [16,17]. Moreover, the evaluation of a platelet RNA-based blood test improves the early diagnosis and clinical outcomes of PAH patients [18]. Early diagnosis is also crucial in the context of the prevention of chronic supraventricular tachyarrhythmias, such as atrial fibrillation (AF) and life-threatening ventricular fibrillation (VF). These arrhythmias develop in the conditions of systemic inflammation and oxidative stress, associated with hypertension [5,13,19,20,21], that induce cardiomyocyte and extracellular matrix (ECM) remodeling, which are the dominant cardiac pathological characteristics in PAH.

Myocardial structural remodeling (i.e., hypertrophy and fibrosis) is considered a crucial factor in the occurrence of cardiac arrhythmias as well as mechanical dysfunction in PAH [12,22,23] and in essential hypertension (HTN) [19]. As shown in Figure 1, the response of LV to HTN demonstrates the heterogeneity of the subcellular alterations of the cardiomyocytes and their junctions that underlie disorders in electrical and mechanical coupling. These changes, accompanied by gap junctions (GJ) Cx43 (GJCx43) abnormalities (Figure 2), contribute to the electrical and mechanical dysfunctions and the myocardial instabilities that promote the development of cardiac arrhythmias and HF.

In this context, it should be emphasized that myocardial structural remodeling and subcellular alterations are always accompanied by disorders of electrical coupling protein connexin43 (Cx43) as illustrated in Figure 2.

Hypertrophy of cardiac myocytes and increased collagen deposition in ECM are linked with the downregulation and abnormal topology of Cx43 [19,25,26,27,28], which is highly pro-arrhythmic. Redistribution of GJCx43 from the ID of hypertrophied cardiomyocytes to the lateral sides (Figure 3B) and their prominent disorder in areas of fibrosis (Figure 3C) promote the occurrence of reentry arrhythmias in HTN, and most likely in PAH, as indicated by the changes demonstrated in Figure 4 and Figure 5.

Such changes facilitate electrical uncoupling of cardiac myocytes and alterations of electrical signal propagation, resulting in myocardial electrical instability that in turn promotes atrial or ventricular arrhythmias [24,29]. However, in contrast with HTN, much less is known about cardiac Cx43 alterations in PAH. Nevertheless, available data suggest the pathogenic implication of Cx43 not only in the heart muscle but also in pulmonary vessels [28,30,31].

According to a statement of the recent ESC 2023 congress, the effective treatment of PAH and arrhythmias are challenges in cardiac care. It seems necessary to comprehensively investigate the inter-organ and inter-cellular communication in PAH-induced RV failure and cardiac arrhythmias in patients and animal models [32,33].

This approach should include investigation of gap junction Cx43 (GJCx43) channels and Cx43 hemichannels for the elucidation of their impact on pathogenesis and arrhythmogenesis in PAH. In line with these ideas, the intention of this review is to provide comprehensive information from the better explored HTN and from the, as-yet scarce, information from the less explored PAH disease, in order to stimulate further research to reveal novel therapeutic tools. Protection from oxidative stress and inflammation related to Cx43 (i.e., connexome) and ECM disorders seems to be crucial. Compounds that attenuate inflammation and oxidative stress, such as melatonin, omega-3 fatty acids, SGLT2i, and statins, exhibit antiarrhythmic/cardioprotective properties. Further investigation is required for the exploration of the potential antiarrhythmic benefit associated with the inhibition of Cx43-hemichannel-mediated NLRP3 inflammasome signaling in PAH. Preventing Cx43 hemichannel opening and preserving GJCx43 function will be key for the further research and development of new connexin-based approaches for the in-clinic treatment of hypertensive heart disease. 

## 2. Electrical Instability and Incidence of Cardiac Arrhythmias in PAH

Patients suffering from PAH are at increased risk of developing of cardiac arrhythmias, [11,34,35,36,37], which are serious complications. The incidence of supraventricular and ventricular tachycardias associated with adverse outcomes has been reported in 8% to 38% of PAH patients [11,14]. During the 3-year follow-up period, 1/3 of patients with PAH developed supraventricular arrhythmias, which were related to the worsening of hemodynamic and functional parameters and which independently predicted adverse prognosis [38,39]. Patients with atrial arrhythmias exhibited higher right atrial (RA) pressure, pulmonary wedge pressure, NT-proBNP and thyroid disease prevalence as well as higher mortality. Atrial remodeling in PAH patients contributes to a higher incidence of supraventricular arrhythmias [40], which are treated by pharmacological or electrical cardioversion and radiofrequency ablation. AF has been shown to be prevalent in 31.1%. in patients with PAH [41]. 

Of note, RA function is associated with changes in RV function and there is RA–RV interaction in PAH [42]. RA–RV uncoupling is evident in PAH patients with atrial fibrillation (AF). Atrial flutter and AF develop in a sizable number of patients [15,41,43], most likely due to fibrosis-related electrical conduction abnormalities [44]. The incidence of AF tremendously worsens cardiac symptoms [40,45]. Therefore, it is essential in PAH to control heart rhythm. 

Premature ventricular beats are more frequent in those subjects with higher adrenergic drive and lower oxygen saturation, while patients with episodes of syncope exhibit a relatively higher vagal activity [46]. 

ECG alterations [46,47] and electrical instability are important predictive factors of life-threatening events in patients with pressure overload of either LV or RV [48]. Moreover, right–left heart interactions and electro-mechanical interactions may be helpful when using ECG as an electrophysiological imaging technology [48]. Electrophysiological changes can facilitate the recognition of pathophysiological processes in the heart. 

ECG has revealed ST segment depression and T wave inversions affecting repolarization in PAH, which might be useful for diagnostics. Moreover, QTc dispersion and prolonged QT/QTc interval positively correlated with pulmonary arterial pressure and were seen to be significantly increased in patients with severe PAH [49]. P-wave dispersion might be an effective ECG indicator for PAH patients for assisting early diagnosis, disease severity assessment and prognosis evaluation [50]. Prolonged QRS duration was seen to be a predictive factor for ventricular arrhythmias that were increased in chronic RV volume overload [51]. 

In this context, it is interesting to note that pressure overload-induced RV failure was shown to be associated with electrophysiological remodeling of the atrophic LV [52] and contractile dysfunction [53,54]. Longer action potentials (AP) and conduction slowing were observed, due to a 24% reduction of Cx43 levels that impaired electrical impulse transmission. 

Alterations in Ca2+handling proteins contribute to RV diastolic dysfunction due to insufficient diastolic Ca2+ clearance [55], which is known to be pro-arrhythmic [56,57,58].

Structural remodeling is always linked with electrical remodeling, mostly due to Cx43 disorders and defective electrical signal propagation among the cardiomyocytes that increase arrhythmogenesis [19,59,60]. Indeed, the heterogeneous expression of Cx43 in the myocardium of the right ventricular outflow tract may promote its dysfunction and serve as substrate for idiopathic ventricular arrhythmia [61]. Available data suggest that the impairment of intercellular electrical coupling and signaling via Cx43 channels may be involved in both the pathomechanisms underlying PAH [28,62,63] and the occurrence of cardiac arrhythmias in PAH, as in primary HTN [19,26,64,65,66]. 

In an animal PAH model, an abnormal and proarrhythmic topology of Cx43 on the lateral sides of cardiac myocytes in RV was detected [28,67]. The disorganization of Cx43 became more pronounced with the progression of hypertrophy and fibrosis, while the proportion of Cx43 at the intercalated disk progressively decreased in PAH [62]. In parallel, conduction velocity and anisotropic ratio in RV were significantly lower than in control rats [28,62]. These conditions promote the development of arrhythmias. However, current therapies aiming to specifically attenuate RV remodeling and improve RV function in PAH [68] did not pay attention to the ECG analysis that might suggest Cx43 disorders, as in HTN [59]. 

Taken together, cardiac arrhythmias, AF and VF, are increasingly recognized as serious complications in PAH contributing to symptoms, morbidity, mortality, and sudden cardiac death. However, there is still a paucity of epidemiological, pathophysiological, and outcome data to guide management of these patients. Undoubtedly, more attention should be paid to systemic inflammation and oxidative stress, relevant factors that promote development of PAH and arrhythmogenesis.

## 3. Factors and Mechanisms Involved in the Occurrence of Cardiac Arrhythmias: Cx43 as a Key Player

In the heart the electrical activation of the pacemaker cells in the sinoatrial node (SA) is conducted through the atria, atrioventricular node (AV) and via the Purkinje conduction system into the ventricles [69,70]. Along this pathway, all cardiac myocytes are activated by currents that flow through gap junction (GJ) Cx channels. In the heart, three main GJ channel proteins are expressed, Cx40 in the atria, dominant Cx43 in the atria and ventricles and Cx45 in the SA and AV nodes as well as in the ventricular conduction Purkinje system [69,70]. Distribution of the action potential and coordinated electrical activation of the heart is maintained by the coupling of atrial and ventricular myocytes via phosphorylated GJCx43 channels at the intercalated disks [71]. Cardiac GJCx43 channels ensure the propagation of both the electrical and the molecular signals that are essential for myocardial synchronization and proper heart function [69,71]. Notably, there are no differences in Cx43 expression between RV and LV in either animals or humans [62].

It is generally accepted that the main factors involved in the development of severe cardiac arrhythmias consist of arrhythmogenic structural substrate (hypertrophy and fibrosis), ectopic triggers (early after-depolarization (EAD) and delayed after-depolarization (DAD)) and modulating elements, such as the autonomic nervous system, humoral elements (e.g., renin-angiotensin-aldosterone system), redox status, inflammation and ischemia [72,73,74,75,76,77,78]. These conditions underly the electrophysiological mechanisms of arrhythmias, which include aberrant impulse formation due to triggered activity or the enhanced automaticity and slowing of impulse conduction promoting re-entrant excitation [73,74,79].

Comprehensive research indicates that Cx43 may be implicated in all proarrhythmic processes in hypertension affected heart as it is illustrated in Figure 2 and Figure 6. Reduced Cx43 expression and its abnormal topology on the lateral sides of hypertrophied cardiac myocytes, as well as their disordered distribution in the fibrotic ventricles of rodents with HTN [19,26,80,81] or with PAH [28,62,72,82,83,84], may underly arrhythmogenic setting [27], which promotes non-uniform anisotropy, conduction defects and re-entry [85,86,87,88], as well as ventricular mechanical dysfunction [89,90]. Abnormal Ca2+handling and Ca2+ overload [56,57,58,91], as well as acidosis (due to ischemia or insufficient perfusion [75]), may induce alterations in the phosphorylation and dephosphorylation of GJCx43 channels with reduced permeability and even electrical uncoupling [29,92,93,94], thereby promoting the triggered ectopic excitation and conduction slowing [95]. The functional remodeling of Cx43 occurs by the regulation of Cx43 phosphorylation that impacts arrhythmogenesis [94,96,97] in HTN [19,26]. Alterations in autonomic tone, sympathetic vs. parasympathetic activity, and humoral factors including RAAS are considered modulating elements regarding susceptibility of the heart to arrhythmias [71,74]. 

In addition, it should be taken into consideration that electrical coupling via GJCx43 channels depends on the mechanical coupling provided by adherence junctions (AJ) and desmosomes (D) at the intercalated disk [98,99,100].The integrity of these structures, connected with GJCx43 and defined as a “connexome” [19], is deteriorated in HTN and most likely in PAH due to collagen deposition, fibrosis and Ca2+ overload [56,58,64], thereby increasing cardiac arrhythmia susceptibility [19,101]. Cell adhesion molecules are also critical in fibrotic progression [102]. 

The pro-arrhythmic signaling of systemic and tissue inflammation [5,13,103,104,105,106], as well as oxidative stress [5,77,107,108,109], resulting in the downregulation of Cx43 and the deterioration of GJCx43-channel-mediated intermyocyte communication in hypertensive heart diseases [19,20] should be emphasized. Notably, Cx43 hemichannels in the cardiac and vascular systems are involved in NOD-like receptor protein-3 (NLRP3) inflammasome signaling [24,110]. It is most likely that Cx43 hemichannels are activated in HTN as well as in PAH [111] and, along with lateralized GJCx43 channels, contribute to arrhythmogenesis [20,24]. NLRP3, via Cx43 hemichannels, also promotes aberrant and pro-arrhythmic diastolic Ca2+ leak and triggered ectopic activity (DAD or EAD) [112]. AF has been shown to be promoted by the enhanced activity of the NLRP3 inflammasome in atrial cardiac myocytes [106]. 

Interestingly, heterogeneous Cx43 expression in the RV outflow tract is considered substrate for idiopathic ventricular arrhythmias [61]. Moreover, differences between LV and RV electrophysiology during pathophysiological remodeling [79,113,114] may enhance arrhythmogenicity. Predominant RV electrical remodeling promotes multiwavelet re-entry which underlies ventricular tachycardia [83]. In this context, the strong proarrhythmic impact of myocardial extracellular matrix (ECM) remodeling should be emphasized, including the fibrosis in PAH [11,12,22,23,115] and HTN [19,116] that is promoted by neurohumoral factors. ECM remodeling is associated with electrical instability due to impairment and/or loss of the GJCx43-channel-mediated electrical coupling of cardiac myocytes [19,117,118,119] that is essential for AP propagation. Accordingly, fibrosis contributes to both mechanical heart failure and the occurrence of malignant re-entrant ventricular arrhythmias VT or VF [90], as well as persistent supraventricular arrhythmias in the setting of high atrial and ventricular pressure [14,120,121]. Moreover, there is the clinical impact of cardiac fibrosis on arrhythmia recurrence after ablation of triggers [122]. In turn, inhibition of pro-fibrotic TGF-β1 signaling [123] and preservation of Cx43 via the prevention/attenuation of inflammation and oxidative stress appear to be a promising therapeutic strategy in PAH, as it is in HTN [19,20]. Notably, myocardial fibrosis and diastolic dysfunction are reversible in hypertensive heart disease, in response to pharmacological intervention with lisinopril [124] and perhaps with other drugs. This issue requires more attention and further research. 

The abovementioned stressful factors may deteriorate the integrity of ID, accompanied with the reduction and lateralization of Cx43 [100,125]. The altered topology of GJCx43, along with the disruption of the adherens junctions and desmosome may result in conduction slowing [126] as well as in electro-mechanical disorders. Therefore, it can be expected that the preservation of GJCx43 channels and the improvement of cardiac-GJCx43-mediated communication [97,103,127,128], as well as the integrity of the intercalated disks [58], could be a promising antiarrhythmic strategy. 

Taken together, pathogenesis of PAH, as with essential HTN, is influenced by genetic, epigenetic, and environmental factors. Arterial hypertension is a common causative factor of vascular as well as cardiac remodeling and dysfunction [33,129]. Given that NLRP3 inflammasome is a key driver of vascular disease [130] and heart failure [131], it appears that enhanced NLRP3 signaling via Cx43 hemichannels may be implicated in pulmonary artery remodeling and endothelial, as well as RV, dysfunction. Indeed, inhibition of the NLRP3 inflammasome by melatonin has been shown to have alleviated acute lung injury [104]. This suggests that Cx43 abundantly expressed in the heart, lung and vessels might be a promising therapeutic and antiarrhythmic target in PAH [30,62,63,110,115,132]. Indeed, Cx43 mimetic peptides [133] (e.g., Gap26, Gap27 and Peptide5) have been reported as therapeutic candidates for the disease processes linked to aberrant Cx43 and some have advanced to clinical testing in humans [134].

## 4. Progress in Research and Treatment in PAH with the Potential to Prevent Cardiac Arrhythmias

Investigation and cardio protection of the RV are less established [135,136], whereas the molecular mechanisms of conditioning in the LV are well characterized [62,113]. From the perspectives of novel therapeutic strategies for right heart ventricle failure and the prevention of cardiac arrhythmias in PAH, the lesson from the left heart seems to be relevant [19,137,138].

The primary pathomechanisms in PAH appear to be inflammation and oxidative stress, which affects various cell types, such as endothelial cells, smooth muscle cells, pericytes and fibroblast, as well as inflammatory cells and platelets [13,139,140,141]. This suggests that PAH patients may benefit from multitargeted therapy [5,142] that focuses on the improvement of vascular function, along with afterload reduction [3,137,143].

Vascular dysfunction is crucial in the pathophysiology of PAH [32] as well as in HTN (Tomiyama 2023). The NLRP3 inflammasome [130,144] is one of the key drivers. Inflammasome signaling is transmitted via the Cx43 hemichannels and Pannexin1 channels [145,146] that promote pro-inflammatory and pro-fibrotic processes [24]. 

Melatonin (a pineal hormone), via inhibition of the inflammasome-associated activation of endothelium and macrophages, attenuates PAH [105,147] and most likely arrhythmogenesis. Indeed, melatonin has been shown to reduce susceptibility of HTN rat heart to VF that was associated with increased Cx43 expression in LV [80]. In addition, melatonin, as a potent antioxidant, attenuates the abnormal proarrhythmic topology of Cx43 and suppresses fibrosis in catecholamine-stressed HTN rats [81]. These findings provide a basis for the application of melatonin that is clinically focused on inflammasomes and reactive oxygen species (ROS) as a possible target of PAH treatment. This may include mitochondrial uncoupling proteins involved in the restriction of ROS production [148]. 

### 4.1. Benefits of SGLT2i Therapy

Preclinical and clinical studies [149] indicate that sodium glucose co-transporter-2 inhibitors (SGLT2i) attenuate endothelial and microvascular dysfunction via several interplaying molecular mechanisms linked with the suppression of inflammation and oxidative stress resulting in vasodilation and beneficial cardiovascular effects. Indeed, SGLT2i exerts direct anti-inflammatory and anti-oxidative effects that ameliorate endothelial dysfunction [150], one of the main pathomechanisms in PAH. Targeting inflammation via SGLT2i canagliflozin may prevent vascular calcification [151,152] and suppress fibrogenesis by empagliflozin [153]. Dapagliflozin has been shown to reduce the risk of severe ventricular arrhythmias in patients with HF [154]. Empagliflozin protects the heart against experimental ischemia/reperfusion-induced SCD via activation of the ERK1/2-dependent cell-survival signaling pathway [155]. Notably, dapagliflozin attenuated vulnerability to arrhythmias by regulating Cx43 expression and enhancing its phosphorylation via the AMPK pathway in post-infarcted rat hearts [156]. 

Emerging evidence on the ability of SGLT2i to modify epigenetic signatures in cardiovascular diseases has stimulated the investigation of a possible implication of these drugs in the development of cardiac arrhythmias [157].

Empagliflozin suppresses the production of mitochondrial reactive oxygen species and mitigates the inducibility of AF [158]. Furthermore, empagliflozin has been shown to attenuate fibrosis and the downregulation of Cx43 as well as to shorten QT interval in mice with metabolic syndrome [159]. Empagliflozin has also been shown to suppress cardiac fibrogenesis through the inhibition of a sodium–hydrogen exchanger and modulation of the calcium homeostasis in human fibroblasts [153]. Pretreatment with empagliflozin has been shown to protect the heart from the lethal ventricular arrhythmia induced by myocardial ischemia and reperfusion injury. These protective benefits may occur because of the activation of the ERK1/2-dependent cell-survival signaling pathway in a glucose-independent manner [155].

Dapagliflozin has been shown to reduce the risk of serious ventricular arrhythmia, cardiac arrest, or SCD when included in the conventional therapy offered to patients with HF, with reduced ejection fraction [154]. Reduced ventricular ectopic burden suggests an early antiarrhythmic benefit of dapagliflozin in patients with HF [160] that might be associated with significantly reduced SCD and death from progressive HF [161].

### 4.2. Targeted Treatment

The multitargeted benefits of sotatercept [162] include various vascular actions and anti-remodeling effects, associated with the inhibition of Smad2/3 activation and downstream transcriptional activity. In this context it would be interesting to explore the effect of sotatercept on cardiac fibrosis, Cx43 expression and topology, and the vulnerability of the heart to arrhythmias. 

Targeting upregulated Wnt-β Catenin-FOSL signaling in PAH using pharmacological inhibition with porcupine O-acyltransferase ameliorated the RV remodeling and collagen deposition [68] that may influence Cx43 expression/distribution and arrhythmogenesis. 

Recent findings have identified the upregulation of long noncoding RNA H19 in decompensated RV in PAH patients that correlated with RV hypertrophy and fibrosis [163]. Therefore, RNA H19 is suggested as a new therapeutic target and a promising biomarker of PAH severity and prognosis.

Augmentation of ACE2 and conversion of angiotensin II to angiotensin-(1–7) improved PAH in rodent [164] and reduced markers of oxidant and inflammatory mediators in human PAH [164]. The anti-fibrotic potential of angiotensin (1–7) has been reported in hemodynamically overloaded rat heart [165]. 

Trandolapril and losartan attenuated pressure and volume overload-induced adverse alterations of cardiac Cx43 and ECM in hypertensive Ren 2 transgenic rats [166]. Pirfenidone inhibited the production of tTGF-β1 and diminished fibroblast proliferation and collagen production [167]. 

The antiarrhythmic effect of ranolazine has been associated with the inhibition of the late sodium current and the suppression of RV remodeling and fibrosis. These results support the notion that ranolazine can improve the electrical and functional properties of the right ventricle, highlighting its potential benefits in the setting of RV impairment [34]. 

Nintedanib, a tyrosine kinase inhibitor that has been approved for the treatment of idiopathic pulmonary fibrosis has been shown to be favorable to RV remodeling due to the inhibition of cardiac fibroblast activation, decreased RV dilatation and reduced hypertrophy [168].

A benefit of the β3 adrenergic agonist mirabegron was shown to be a significant improvement in RV ejection fraction in PAH assessed during the SPHERE-HF trial [169]. 

Stimulating the parasympathetic activity of pyridostigmine through the inhibition of acetylcholinesterase improved survival, RV function, and pulmonary vascular remodeling in experimental PAH [170].

Detrimental neurohormonal overactivation was inhibited by renal denervation that reduced pulmonary vascular remodeling in experimental PAH [171].

The epigenetic mechanisms involved in the pathogenesis of PAH, specifically DNA methylation, histone modifications, and microRNAs may be a therapeutic potential [18,172]. Bromodomain and extra-terminal domain proteins are epigenetic modulators and bromodomain-containing protein 4 (BRD4) is involved in the inflammation of several major human lung diseases [173]. Preclinical findings suggest the benefit of targeting these proteins using pharmaceutical inhibitors in different lung diseases.

In the context of the prevention or attenuation of myocardial fibrosis, more attention should be paid to non-coding RNAs, such as microRNA-21, microRNA-29 and long RNAH19, that have become not only biomarkers but could also serve as therapeutic targets with which to fight fibrosis [115,163]. Indeed, inhibition of microRNA-206 ameliorates arrhythmias via targeting Cx43 [174].

According to the 2022 ESC/ERS guidelines for the diagnosis and treatment of PAH [3], recommendation therapies include endothelin receptor antagonists, phosphodiesterase type 5 inhibitors, soluble guanylate cyclase stimulators, prostacyclin analogues and prostacyclin receptor agonists. Improvement of vascular function, along with afterload reduction, is still the cornerstone in the treatment of PAH [132,137,143,175,176]. Progress in imaging techniques, such as positron emission tomography, may become helpful in the determination of neurohormonal status in PAH patients of different disease stages and the optimization of individual treatment responses [177]. 

Finally, it should be noted that the task of ERS is to increase the availability and awareness of exercise training and rehabilitation programs for PAH patients as an important treatment option that has been shown to be beneficial in cardiovascular diseases [178]. Endurance exercise training in patients with stable PAH has a positive effect, promoting potential mechanisms of damage repair. This effect could contribute to a positive hemodynamic and clinical response [179], as with HTN [180]. 

Moreover, nutritional deficiencies, such as those of iron, vitamins D, B12, K1 and selenium [181], have been found in patients with PAH. The improvement of dietary intake and future research to demonstrate the clinical importance of nutritional interventions are ongoing tasks. 

Altogether, nutrition and lifestyle interventions may not only improve the quality of life of patients with PAH [181,182], but most likely also prevent development of the disease.

## 5. Conclusions

Although advances in PAH therapy have improved outcomes, poor survival remains worldwide [183]. There are still gaps in our knowledge of arrhythmias in PAH in terms of both pathogenesis and optimal management strategies [184]. One of the most important issues in PAH is late diagnosis, since screening or diagnostic efforts are often overlooked due to the rarity of the disease. Currently there are no selective treatments targeting the failing right ventricle. New treatments directly targeting the crucial pathological determinants of RV failure and arrhythmogenesis are still emerging. Moreover, the discovery of novel disease pathways and modifiers affecting the pulmonary circulation requires intense investigation [185].

Due to the health benefits conferred by the early detection of PAH, as well as the identification of novel PAH-associated genes and biomarkers, [186] perspectives seem to be better [187]. The length of survival has improved with recent advances in specific therapy, although it is still the case that only 65% to 70% of patients survive five years with a PAH diagnosis [11]. Unfortunately, there are no studies that definitively confirm that specific PAH therapies reduce the risk of cardiac arrhythmias. Further research is also essential to elucidate sex differences in the development of HTN and PAH for efficient treatment response [19]. 

Endothelial dysfunction in arterial hypertension as a complex interplay has been a matter of study in recent years as a strategy while treating hypertension [137]. Considering NLRP3 inflammasome as a key driver of vascular disease and endothelial dysfunction [130] suggests the implication of Cx43 hemichannels, which might be therapeutic targets. The potential benefit of the inhibition of Cx43-hemichannel-mediated NLRP3 inflammasome signaling in PAH requires further investigation. This challenging strategy may attenuate or reverse the process of myocardial fibrosis and the downregulation and mislocalization of GJCx43, thereby offering protection from malignant arrhythmias in hypertension. Moreover, a peptide mimetic of the Cx43 carboxyl-terminus reduces GJ remodeling and the incidence of arrhythmia [133].

Numerous experimental findings and results of clinical trials undoubtedly indicate the use of a multitargeted approach to prevent or attenuate the development of malignant arrhythmias in pathophysiological conditions, including PAH. Suppression of ECM remodeling and the preservation of the cardiac GJCx43 channel’s function and topology, for proper electrical signal propagation supporting myocardial electrical stability, is “conditio sine qua non”.

Thus, clinicians should monitor the markers of inflammation and oxidative stress in order to reduce their adverse effects in hypertensive patients. This view is supported by literature selection process outlined in Figure 7. 

## Figures and Tables

**Figure 1 ijms-25-03275-f001:**
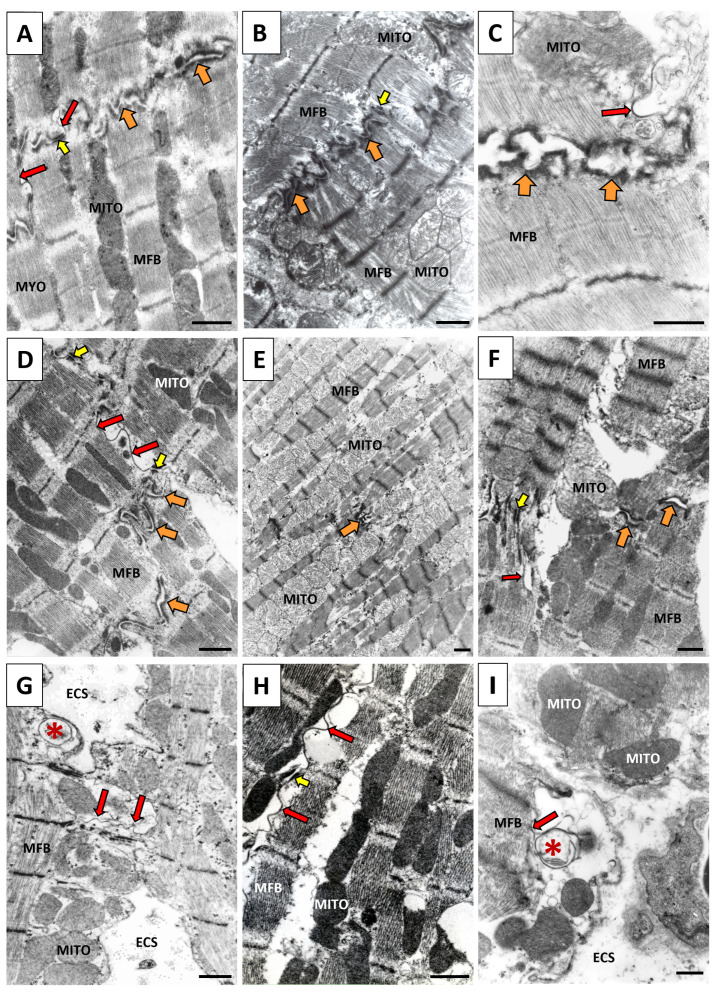
Representative electron microscope images of cardiomyocytes from the left ventricle of a hypertensive rodent heart. Note the apparent heterogeneity of the subcellular alterations (**D**–**I**) in response to HTN, which may also be expected in the response to PAH. (**A**) Cardiomyocytes are connected via the compact structure of the intercalated disk composed of GJCx43 (red arrows), adherens junctions (orange arrows) and desmosome (yellow arrows) in the healthy heart. (**B**) Asynchrony of contraction between neighboring cardiomyocytes due to electrical uncoupling of GJCx43. (**C**) Impairment of cardiomyocyte adhesion due to the dehiscence of adherens junctions in the vicinity of GJCx43. (**D**) Hypertrophied cardiomyocytes coupled with laterally located GJCx43. (**E**) Ischemic cardiomyocytes connected with rudimentary adherens junctions. (**F**) Hypercontracted cardiomyocytes (left corner) due to Ca^2+^ handling disorders are connected with relaxed cardiomyocytes (right corner), demonstrating the asynchrony of contraction and the involvement GJCx43. (**G**) Internalization (star) and destruction of lateral GJCx43 due to the pronounced extracellular space remodeling. (**H**) Long lateral GJCx43 connecting hypertrophied cardiomyocytes. (**I**) Myocardial interstitial fibrosis associated with the widening of extracellular space and internalization of GJCx43 (red star), which had undergone proteasome degradation. Mito—mitochondria, MFB—myofibrils, ESC—extracellular space. Scale bar: 0.5 μm. Adapted from [19,24].

**Figure 2 ijms-25-03275-f002:**
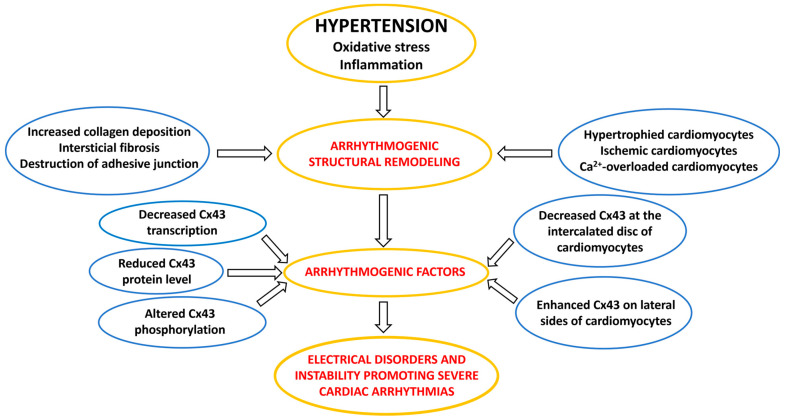
A flow chart illustrating the arrhythmogenic factors involved in the development of cardiac arrhythmias in hypertension.

**Figure 3 ijms-25-03275-f003:**
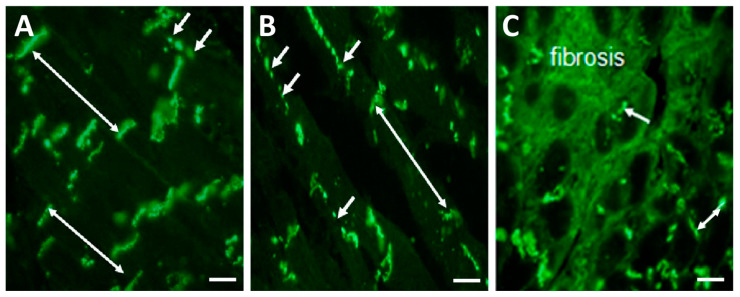
Representative immunofluorescence images of GJCx43 topology in the left rat heart ventricles. (**A**) Dominant polar localization (double arrows) of GJCx43 at the ID and sporadically on lateral sides of cardiomyocytes (short arrows) in healthy rat heart. (**B**) Enhanced lateral localization of GJCx43 (short arrows) and reduced at the ID (double arrow) in hypertrophied cardiomyocytes. (**C**) Reduced and prominently disordered GJCx43 in areas of fibrosis (short arrow and double arrow). Scale bar: 100 μm. Unpublished images.

**Figure 4 ijms-25-03275-f004:**
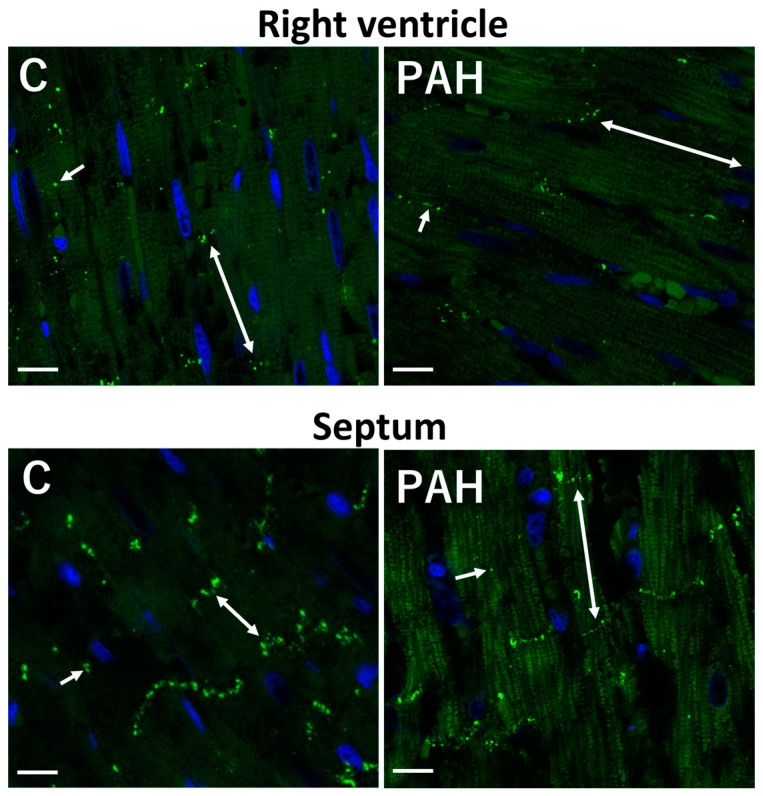
Reduced immunolabeling of GJCx43 in the right ventricle and septum in monocrotaline-induced PAH in rat. Polar localization (double arrows) of GJCx43 at the ID and sporadically on lateral sides of cardiomyocytes (short arrows). Scale bar: 15 μm. Unpublished confocal images.

**Figure 5 ijms-25-03275-f005:**
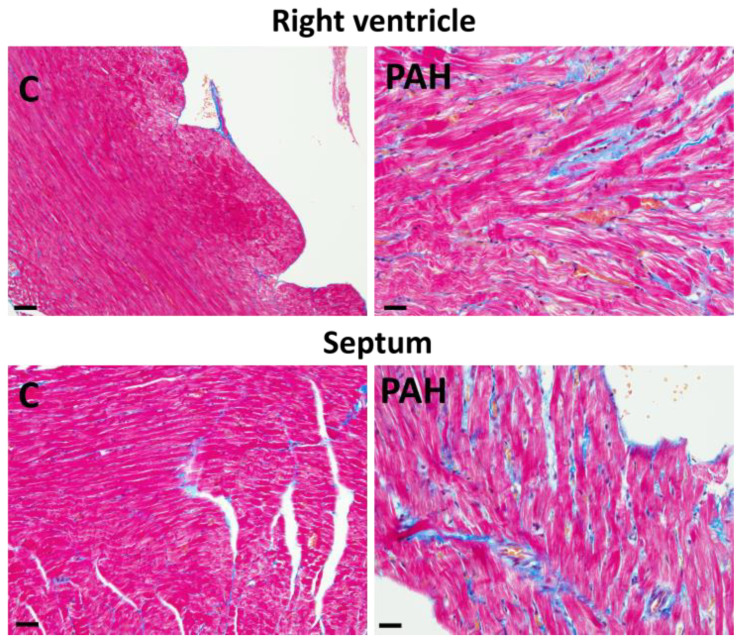
Interstitial fibrosis (blue color) in the right heart ventricle and septum in monocrotaline-induced PAH in rat. Masson trichrome staining. Scale bar: 15 μm. Unpublished light microscopic images.

**Figure 6 ijms-25-03275-f006:**
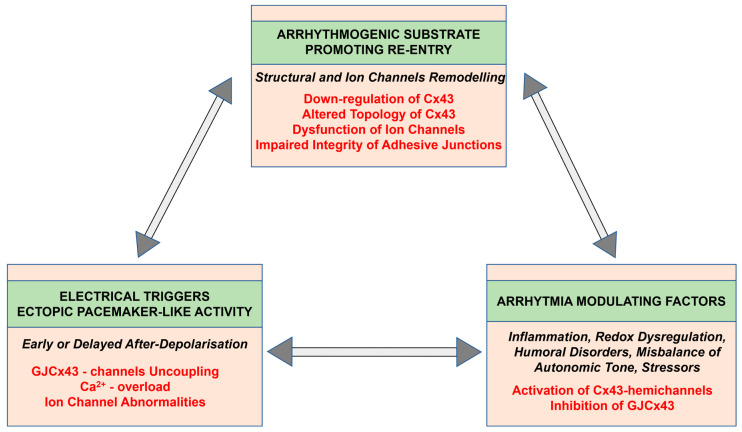
Potential mechanisms of cardiac arrhythmia generation and implication of connexin-43.

**Figure 7 ijms-25-03275-f007:**
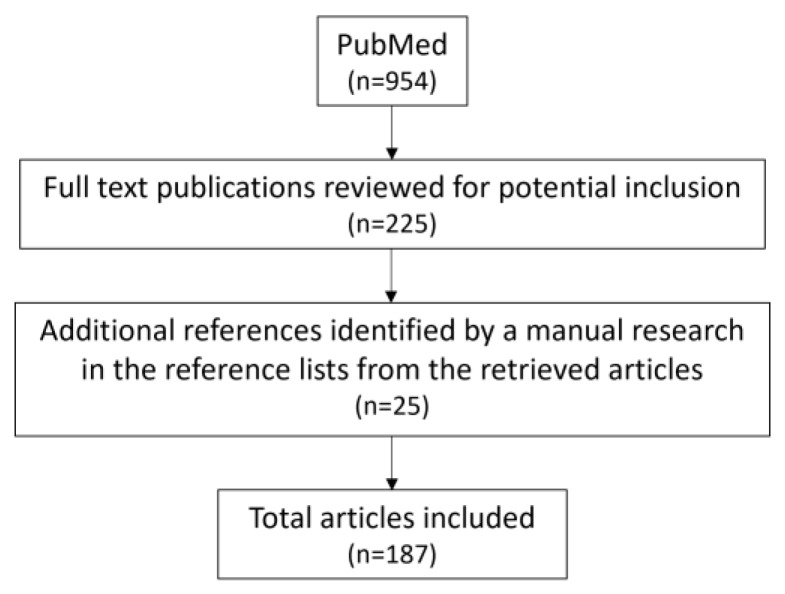
Flow chart of the literature selection process for the present article.

## Data Availability

Data is contained within the article.

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
