# Peer review of "Connexin43, A Promising Target to Reduce Cardiac Arrhythmia Burden in Pulmonary Arterial Hypertension"

_ijms, 2024, doi:10.3390/ijms25063275_

Round 1
Reviewer 1 Report
Comments and Suggestions for Authors
The topic is interesting and the paper is quite well written. Nevertheless, in my opinion, some parts need to be improved, I have some comments:
1) Abstract. Treatment algorithm in HTN is superior to 23 PAH likely due to paucity of comprehensive pathomechanisms and causal factors for multitargeted 24 approach in PAH. The intention of this review is to provide available information related to cardiac 25 arrhythmias focusing on Cx43 and effects of therapy in PAH. Discovering novel therapeutic and 26 antiarrhythmic strategies may be challenging for further research. Abstract might be beneficial to include a sentence that briefly summarizes the key findings of the study. This can provide readers with a quick overview of the research.
2) Myocardial structural remodeling (i.e. hypertrophy and fibrosis) is considered as a 63 crucial factor in the occurrence of cardiac arrhythmias as well as mechanical dysfunction 64 in PAH [12,21,22] likewise in essential hypertension (HTN) [18]. As shown in Figure 1, the 65 response of LV to HTN points out heterogeneity of subcellular alterations of the cardio- 66 myocytes and their junctions that underlie disorders in electrical and mechanical cou- 67 pling. These changes accompanied by Cx43GJ abnormalities (Figure 2.) contribute to elec- 68 trical as well as mechanical dysfunction and myocardial instability promoting develop- 69 ment of cardiac arrhythmias and HF.
This approach should include 120 investigation of gap junction Cx43 (GJCx43) channels and Cx43-hemichannels to elucidate 121 their impact on pathogenesis and arrhythmogenesis of PAH. In line with these ideas the 122 intention of this review is to provide relevant information that may stimulate further re- 123 search and novel therapeutic approaches.
Please, improve the description of the aim of the study and the underline the novelty of this observation.
3) Please re-write the introduction the paragraph is not clear
4) Please add in the introduction a brief part on the methods used to evaluate the papers to mede the review.
5) Please, add in the introduction a flowchart to clarify the revision precess
6) Please, add a brief summary in the introduction to present the other paragraphs of the paper
7) Conclusions. Please, underline the novelty of the paper and the possible clinical implications.
Comments on the Quality of English Language
Minor changes of English language are required
Author Response
Thank you very much for reviewing our manuscript and relevant comments and suggestions. We did our best to follow your thoughts hopping that our revision contributes to improvement of this review article.
We would like to note that all information included in this review article were obtained from PubMed data base with prevalent clinical papers dealing with PAH and much less experimental data from animal models. Therefore, we aimed to discuss this serious clinical issue to support further innovative research focusing on key factors in pathomechanisms of PAH as well as cardiac arrhythmias, i.e. chronic inflammation and oxidative stress. These factors result in cardiac disorders and dysfunction of the key protein involved in electrical coupling, connexin43. Comprehensive data indicated its downregulation and alteration in topology is not only pro-arrhythmic but also pro-fibrotic.
The topic is interesting and the paper is quite well written. Nevertheless, in my opinion, some parts need to be improved, I have some comments:
1) Abstract. Treatment algorithm in HTN is superior to 23 PAH likely due to paucity of comprehensive pathomechanisms and causal factors for multitargeted 24 approach in PAH. The intention of this review is to provide available information related to cardiac 25 arrhythmias focusing on Cx43 and effects of therapy in PAH. Discovering novel therapeutic and 26 antiarrhythmic strategies may be challenging for further research. Abstract might be beneficial to include a sentence that briefly summarizes the key findings of the study. This can provide readers with a quick overview of the research.
We revised as follows: The intention of this review is to provide information focusing on implication of Cx43 in development of cardiac arrhythmias in hypertensive heart disease. Besides, progress in cardioprotective and potentially antiarrhythmic effects of therapy is included. Namely, benefits of SGLT2i, sotatercept, pirfenidone, ranolazine, nintedanib and mirabegron as well as melatonin are discussed. Discovering novel therapeutic and antiarrhythmic strategies may be challenging for further research. Undoubtedly, it should include protection of the heart from inflammation and oxidative stress, as primary pro-arrhythmic factors jeopardizing cardiac Cx43 homeostasis, integrity of intercalated disk and extracellular matrix, thereby heart function.
2) Myocardial structural remodeling (i.e. hypertrophy and fibrosis) is considered as a 63 crucial factor in the occurrence of cardiac arrhythmias as well as mechanical dysfunction 64 in PAH [12,21,22] likewise in essential hypertension (HTN) [18]. As shown in Figure 1, the 65 response of LV to HTN points out heterogeneity of subcellular alterations of the cardio- 66 myocytes and their junctions that underlie disorders in electrical and mechanical cou- 67 pling. These changes accompanied by Cx43GJ abnormalities (Figure 2.) contribute to elec- 68 trical as well as mechanical dysfunction and myocardial instability promoting develop- 69 ment of cardiac arrhythmias and HF.
This approach should include 120 investigation of gap junction Cx43 (GJCx43) channels and Cx43-hemichannels to elucidate 121 their impact on pathogenesis and arrhythmogenesis of PAH. In line with these ideas the 122 intention of this review is to provide relevant information that may stimulate further re- 123 search and novel therapeutic approaches.
Please, improve the description of the aim of the study and the underline the novelty of this observation.
We revised as follows: This approach should include investigation of gap junction Cx43 (GJCx43) channels and Cx43-hemichannels for elucidation their impact on pathogenesis and arrhythmogenesis in PAH. In line with these ideas, the intention of this review is to provide comprehensive information from better explored HTN and yet scarce information from less explored PAH disease that may stimulate further research to reveal novel therapeutic tools in clinic. Protection from oxidative stress and inflammation related to Cx43 (i.e. connexome) and ECM disorders seems to be crucial. Compounds that attenuate inflammation and oxidative stress, such as melatonin, omega-3 fatty acids, SGLT2i, statins exhibit antiarrhythmic/ cardioprotective properties in clinical. Further investigation is required for exploring potential antiarrhythmic benefit associated with inhibition of Cx43 hemichannels mediated NLRP3 inflammasome signaling in PAH. Preventing Cx43 hemichannel opening and preserving GJCx43 function will be key for further research and development of new connexin-based approaches for treating hypertensive heart disease in clinic.
3) Please re-write the introduction the paragraph is not clear
We added at the end of introduction the paragraph above.
4) Please add in the introduction a brief part on the methods used to evaluate the papers to mede the review.
We are not sure if we fully understand this comment, we just used and collected data form PubMed data base. Microscopic images used in the manuscript for better elucidation were obtain from our experimental archive.
5) Please, add in the introduction a flowchart to clarify the revision process
In the introduction part is presented in the Figure 2 a flow chart illustrating arrhythmogenic factors involved in the development of cardiac arrhythmias in hypertension, which can describe main issue of this review.
6) Please, add a brief summary in the introduction to present the other paragraphs of the paper
We apologize for difficulties. We added at the end of introduction a brief summary.
7) Conclusions. Please, underline the novelty of the paper and the possible clinical implications.
We included in conclusion: Potential benefit of inhibition of Cx43 hemichannels mediated NLRP3 inflammasome signaling in PAH requires further investigation. This challenging strategy may attenuate or reverse the process of myocardial fibrosis, downregulation and mislocalization of GJCx43, thereby protect from malignant arrhythmias in arterial hypertension. Moreover, a peptide mimetic of the Cx43 carboxyl-terminus reduces GJ remodeling and arrhythmia following ventricular injury (131). Numerous experimental findings and results of clinical trials undoubtedly point out multitargeted approach to prevent or attenuate development of malignant arrhythmias in pathophysiological conditions including PAH. Suppression of ECM remodeling and preservation of cardiac GJCx43 channel’s function and topology for proper electrical signal propagation supporting myocardial electrical stability is “conditio sine qua non”.
Thus, clinicians should monitor markers of inflammation and oxidative stress to reduce their adverse effects in hypertensive patients.
Author Contributions: Conceptualization, writing review
Comments on the Quality of English Language
Minor changes of English language are required
Submission Date
12 February 2024
Date of this review
15 Feb 2024 10:36:19

Reviewer 2 Report
Comments and Suggestions for Authors
Dear authors , I congratulate you with a well written and quite thorough overview on what is known on the effect of PAH on cardiac remodeling and subsequent arrhythmia risk. Some of yor insightsts/recommendations are quite logical, but so much easier said than done! For example: You propose that therapy targeted at preserving functional CX43 channels and maintaining intercalated disc integrity would be a promising antiarrhythmic strategy. I fully agree, but this is so much easier said than done.. I would have liked the paper much better if the you had shared with us your ideas how to achieve such therapy, even if these ideas are very speculative.
The remainder of my remarks is mostly textual improvements to make your paper even better:
Abstract:
Line 12: unusual grammatical construction: change to for example: Whereas essential hypertension (HTN) is very prevalent, pulmonary arterial hypertension (PAH) is very rare.
Line 23: The treatment algorithm……
Introduction:
Line 51: 30% of what? (missing information!) I presume you mean to say 30% of PAH related deaths?
figure 1: a legend for all the abbreviations in these pictures would be helpful even when most of them are self explanatory! ID?, myo?, ecs? When you say hypertensive rodent, I take it you mean rodent with induced pah and not with systemic hypertension. please be clear about this in the text! Pleasse add in de legend what exactly we are looking at! I suppose sections of rodent RV? please state this in the text.
Line 112: arterial should read atrial?
Ine 152: diagnostic change to diagnosis or diagnostics?
Line176-177: incorrect grammar: aiming to specifically attenuate RV remodeling and improve RV function
Line178: likewise change to like in, or as in
Line193: AP, maybe spell out action potentials for readability.
Line247: likewise change to like in, or as in
Line258-259: Sounds logical, but is much easier said than done, are you familiar with any drugs that could potentially achieve maintenance of intercalated disc integrity, please mention these. So, please elaborate even if your suggestions are very tentative.
You do not mention thisin the text, but I see great pathophysiological overlap with ARVC (Arrythmogenic Right Ventricular Cardiomyopathy) which is primarily a disease of the cardiac desmosome, but with the same mechanoelectrical uncoupling as an underlying arrhythmogenic mechanism, any drug that would inprove RV intercalated disc integrity in PAH would be a potential drug to treat ARVC, I would reckon and vice versa.
Ine261: likewise to is an unusual English construction change to like in or as in
4. Progress in research and treatment in PAH with potential to prevent cardiac arrhythmias
Liine278: heart should read ventricle.
Line289: consider changing attenuated into attenuates
Line313: consider changing suppressed into suppresses, likewise mitigate into mitigates (line314)
Line323: HFrEF please explain abbreviation
Line 351: progressed IPF and PAH often coincide. Please provide a reference for the beneficial effect of Nintedanib on RV remodeling, do you suggest a causal mechanism seperate from the known attenuation of IPF by this drug?
Line362: I was confused for a second as also Bone morphogenic protein4 (BMP4), not to be confused with BRD4 is associated with genetic PAH.
Line383: likewise change to like.
Comments on the Quality of English LanguageA few minor point all of which I adressed already above. You seem rather fond of likewise to instead of like in, or as in. I don't believe likewise can be used in this way in correct English, but I am also not a native speaker.
Author Response
Dear reviewer, thank you so much for reviewing our manuscript and for your encouraging compliment. We appreciate your fundamental comments and professional suggestions. We agree with you that we should pay more attention to extrapolate experimental findings to clinical conditions. Maybe from our revision is much clear that protection of the heart from inflammation and oxidative stress is relevant not only in experimental conditions but in clinic as well. Such approach may attenuate hypertension-induced adverse effects resulting in myocardial remodelling, including fibrosis and intercalated disc and Cx43 disorders. Cardioprotective anti-inflammatory effects of colchicine is well known in clinic. On the molecular levels, inhibition of inflammasome signalling is promising strategy in future, as reviewed by Leybaert et al. 2023. Nevertheless, molecular analysis and mechanisms regarding development and consequences of PAH are scarce, therefore, the intention of our review was to challenge further research. We hope that the message for readers of revised manuscript is more clear.
‘’ A satisfactory answer will never satisfy’’. G.K. Chesterton
Dear authors, I congratulate you with a well written and quite thorough overview on what is known on the effect of PAH on cardiac remodeling and subsequent arrhythmia risk. Some of your insights/recommendations are quite logical, but so much easier said than done! For example: You propose that therapy targeted at preserving functional CX43 channels and maintaining intercalated disc integrity would be a promising antiarrhythmic strategy. I fully agree, but this is so much easier said than done I would have liked the paper much better if the you had shared with us your ideas how to achieve such therapy, even if these ideas are very speculative.
The remainder of my remarks is mostly textual improvements to make your paper even better:
We apologize for these mistakes and thanks to you we all abolished.
Abstract:
Line 12: unusual grammatical construction: change to for example: Whereas essential hypertension (HTN) is very prevalent, pulmonary arterial hypertension (PAH) is very rare.
Revised.
Line 23: The treatment algorithm……
Revised
Introduction:
Line 51: 30% of what? (missing information!) I presume you mean to say 30% of PAH related deaths?
Yes, we revised it.
figure 1: a legend for all the abbreviations in these pictures would be helpful even when most of them are self explanatory! ID?, myo?, ecs? When you say hypertensive rodent, I take it you mean rodent with induced pah and not with systemic hypertension. please be clear about this in the text! Pleasse add in the legend what exactly we are looking at! I suppose sections of rodent RV? please state this in the text.
We add in legend that images were from left ventricle of rodent heart with HTN. Further research should include RV as well. However, we assume that subcellular alterations will be similar.
Line 112: arterial should read atrial?
Revised.
Line 152: diagnostic change to diagnosis or diagnostics?
Revised.
Line176-177: incorrect grammar: aiming to specifically attenuate RV remodeling and improve RV function
Revised.
Line178: likewise change to like in, or as in
Revised
Line193: AP, maybe spell out action potentials for readability.
Revised.
Line247: likewise change to like in, or as in
Revised.
Line258-259: Sounds logical, but is much easier said than done, are you familiar with any drugs that could potentially achieve maintenance of intercalated disc integrity, please mention these. So, please elaborate even if your suggestions are very tentative.
You do not mention this in the text, but I see great pathophysiological overlap with ARVC (Arrhythmogenic Right Ventricular Cardiomyopathy) which is primarily a disease of the cardiac desmosome, but with the same mechanoelectrical uncoupling as an underlying arrhythmogenic mechanism, any drug that would improve RV intercalated disc integrity in PAH would be a potential drug to treat ARVC, I would reckon and vice versa.
We agree with your proposal, however, we think that this issue related to integrity of intercalated disc, including adhesive junctions (adherens junctions and desmosome) and Cx43 gap junctions that contribute to mechano-electrical disorders in hypertension likewise in ARVC and also in senescent heart (Bonda 2016) requires comprehensive approach. Nevertheless, we revised this part.
Ine261: likewise, to is an unusual English construction change to like in or as in
Revised.
- Progress in research and treatment in PAH with potential to prevent cardiac arrhythmias
Liine278: heart should read ventricle.
Line289: consider changing attenuated into attenuates
Line313: consider changing suppressed into suppresses, likewise mitigate into mitigates (line314)
Line323: HFrEF please explain abbreviation
Revised.
Line 351: progressed IPF and PAH often coincide. Please provide a reference for the beneficial effect
of Nintedanib on RV remodeling, do you suggest a causal mechanism seperate from the known attenuation of IPF by this drug?
Revision: Nintedanib, a tyrosine kinase inhibitor that has been approved for the treatment of idiopathic pulmonary fibrosis was favorable on RV remodeling due to inhibition of cardiac fibroblast activation, decreased RV dilatation and reduced hypertrophy [166]. Reported in paper of Rol N. 2019
Line362: I was confused for a second as also Bone morphogenic protein4 (BMP4), not to be confused with BRD4 is associated with genetic PAH.
Revised: Bromodomain and extra-terminal domain proteins are epigenetic modulators and bromodomain-containing protein 4 (BRD4) is involved in inflammation of several major human lung diseases [171]. In contrast, BMP4 is genetic factor involved in PAH.
Line383: likewise change to like.
All revised.
Comments on the Quality of English Language
A few minor point all of which I addressed already above. You seem rather fond of likewise to instead of like in, or as in. I don't believe likewise can be used in this way in correct English, but I am also not a native speaker.
Thank you very much: we appreciate your help, your professional comments and ideas that are challenging for further research.
In the presence of real great man even small people feel great too. G.K. Chesterton
Submission Date
12 February 2024
Date of this review
27 Feb 2024 18:03:38

Reviewer 3 Report
Comments and Suggestions for Authors
Major comments:
- reference 33 is an animal study therefore should be removed from the line 127 (phrase refers patients with PAH)
- at line 166-167 the correct statement should be ,,heterogeneous expression of Cx43 in the right ventricular outflow tract area may promote idiopathic ventricular arrhythmia" (reference 61 is a descriptive study of Cx43 distribution in an animal model and NOT a functional one)
- long discussion and not too relevant for PAH on SGLT2i therapy
- section 4.2 Targeted therapy mixing molecules tried in PAH with molecules used in HF/HTN; additionally information about molecules used in PAH are insufficiently detailed.
Minor comments:
- at line 112 ,,atrial and ventricular arrhythmias" instead of ,,arterial and ventricular arrhythmias"
- lines 118&119 repetition ("It seems necessary to comprehensively investigate the inter-organ the 118 need to investigate comprehensively inter-organ and inter-cellular communication")
Author Response
Thank you very much for reviewing our manuscript and relevant comments and suggestions. We did our best to follow your thoughts hopping that our revision contributes to improvement of this review article.
Major comments:
- reference 33 is an animal study therefore should be removed from the line 127 (phrase refers patients with PAH)
We added in text statement that experiments were also performed in animal studies.
We revised as follows: It seems necessary to comprehensively investigate the inter-organ the need to investigate comprehensively inter-organ and inter-cellular communication in PAH-induced RV failure and cardiac arrhythmias in patients and animal models [32,33].
- at line 166-167 the correct statement should be ,,heterogeneous expression of Cx43 in the right ventricular outflow tract area may promote idiopathic ventricular arrhythmia" (reference 61 is a descriptive study of Cx43 distribution in an animal model and NOT a functional one)
We corrected it in the text. We revised as follows:
Indeed, heterogeneous expression of Cx43 in the myocardium of right ventricular outflow tract may promote its dysfunction and serve as substrate for idiopathic ventricular arrhythmia [61].
- long discussion and not too relevant for PAH on SGLT2i therapy
We depicted in the text, that inflammation and oxidative stress are crucial factors in PAH. It follows that direct anti-inflammatory and anti-oxidative effects of SGLT2i is implicated in PAH therapy.
We revised as follows:
Indeed, SGLT2i exert direct anti-inflammatory and anti-oxidative effects that ameliorate endothelial dysfunction [150], one of the main pathomechanism in PAH.
- section 4.2 Targeted therapy mixing molecules tried in PAH with molecules used in HF/HTN;
There is a similarly mechanism of myocardial structural remodelling and connexin 43 disorders in PAH as well as in HTN resulting in arrhythmias and HF.
additionally information about molecules used in PAH are insufficiently detailed.
We revised as follows:
Antiarrhythmic effect of ranolazine was associated with inhibition of the late sodium current and suppression of RV remodeling and fibrosis. These results support the notion that ranolazine can improve the electrical and functional properties of the right ventricle, highlighting its potential benefits in the setting of RV impairment [34].
Nintedanib, a tyrosine kinase inhibitor that has been approved for the treatment of idiopathic pulmonary fibrosis was favorable on RV remodeling due to inhibition of cardiac fibroblast activation, decreased RV dilatation and reduced hypertrophy [168].
Minor comments:
- at line 112 ,,atrial and ventricular arrhythmias" instead of ,,arterial and ventricular arrhythmias"
We corrected it in the text.
We revised as follows:
Such changes facilitate electrical uncoupling of cardiac myocytes and alterations of electrical signal propagation resulting in myocardial electrical instability promoting atrial or ventricular arrhythmias.
- lines 118&119 repetition ("It seems necessary to comprehensively investigate the inter-organ the 118 need to investigate comprehensively inter-organ and inter-cellular communication")
We corrected it in the text. We revised as follows:
It seems necessary to comprehensively investigate the inter-organ and inter-cellular communication in PAH-induced RV failure and cardiac arrhythmias in patients and animal models

Round 2
Reviewer 1 Report
Comments and Suggestions for Authors
Could you please insert a flowchart to clarify the revision process to select the papers for your review-article?
Comments on the Quality of English LanguageMinor changes of English language are required
Author Response
We added flow chart of the literature selection process.
Reviewer 3 Report
Comments and Suggestions for Authors
Major issues:
- authors insists on the discussion section to describe much widely than needed the effect of various molecules on HTN/HF although their review is dedicated to PAH instead to include data (if this exist) on the effect of this molecules on PAH.
Author Response
Thank you for your comment which we would like to explain.
The intention of this article is to stimulate molecular research in PAH, which is still poor comparing to HTN (as you also noted). According to available data related to cardiac arrhythmogenesis and heart failure in hypertension, connexin43 appears as one of the key player. HTN as well as PAH is promoted by chronic inflammation and oxidative stress resulting in connexin43 disorders. These are associated with myocardial remodelling and arryhmogenesis that increase vulnerability of the heart to malignant arrhythmias.